# Navigating the Aerosolized Frontier: A Comprehensive Review of Bioaerosol Research Post-COVID-19

Chengchen Zhang [1,2], Xiaorong Dai [3], Tedros Gebrezgiabhier [1], Yuan Wang [1], Mengrong Yang [1], Leiping Wang [1], Wei Wang [1], Zun Man [1], Yang Meng [1], Lei Tong [1], Mengmeng He [1], Bin Zhou [4], Jie Zheng [1,*] and Hang Xiao [1,*]

1. Key Laboratory of Urban Environment and Health, Ningbo Observation and Research Station, Institute of Urban Environment, Chinese Academy of Sciences, Xiamen 361021, China; chengchenzhang@iue.ac.cn (C.Z.); tedros@iue.ac.cn (T.G.); mryang@iue.ac.cn (M.Y.); lpwang@iue.ac.cn (L.W.); wwang@iue.ac.cn (W.W.); zman@zju.edu.cn (Z.M.); ymeng@iue.ac.cn (Y.M.); ltong@iue.ac.cn (L.T.); mmhe@iue.ac.cn (M.H.)
2. University of Chinese Academy of Sciences, Beijing 100049, China
3. College of Biological and Environmental Sciences, Zhejiang Wanli University, Ningbo 315100, China; xrdai@zwu.edu.cn
4. Animal Husbandry Technology Promotion and Breeding Livestock and Poultry Monitoring Station of Zhejiang Province, Hangzhou 310020, China; zjxmzb@163.com
* Correspondence: jzheng@iue.ac.cn (J.Z.); hxiao@iue.ac.cn (H.X.)

**Abstract:** In the wake of the COVID-19 pandemic, the scientific community has been galvanized to unravel the enigmatic role of bioaerosols in the transmission of infectious agents. This literature review, anchored in the extensive Web of Science Core Collection database covering the period from 1990 to 2023, utilizes a bibliometric approach to chart the dynamic landscape of bioaerosol research. It meticulously documents the paradigm shifts and burgeoning areas of inquiry that have emerged in the aftermath of the pandemic. This review meticulously maps out the sources and detection strategies of pathogens in a variety of ecosystems. It clearly shows that impaction and filtration sampling methods, followed by colony counting and PCR-based detection techniques, were predominantly used in the scientific works within the previous three decades. It synthesizes the progress and limitations inherent in a range of models for predicting aerosol-mediated pathogen spread and provides a comparative analysis of eDNA technology and traditional analytical techniques for bioaerosols. The accuracy of these detection methods and forecasting models is paramount for the early recognition of transmission risks, which, in turn, paves the way for prompt and effective disease mitigation strategies. By providing a thorough analysis of the historical progression and current state of bioaerosol research, this review illuminates the path ahead, identifying the critical research needs that will drive the field's advancement in the years to come.

**Keywords:** bioaerosol; research trend; COVID-19; sampling methods; detection techniques; mathematical models; eDNA technology

## 1. Introduction

Airborne particles consisting of biological components in solid or liquid patterns forming a stable dispersion system are referred to as bioaerosols. These particles encompass a diverse range of biological aerosols, including viable entities like fungi, bacteria, and archaea, nonviable entities like viruses, allergens, toxins, and pollen, as well as resistant structures like fungal spores and bacterial endospores. They vary in size from 0.01 to 100 μm, covering a broad spectrum of sizes [1]. When inhaled, bioaerosols within the 5 to 30 μm particle size range are typically deposited to settle in the oral cavity and respiratory tract, thereby increasing the risk of diseases like tooth decay, gum infection, bronchitis, bronchiectasis, and capillary bronchitis. These particles, characterized by their relatively larger particle size, have a greater tendency to settle in the respiratory tract, thereby increasing the risk of diseases like bronchitis, bronchiectasis, and capillary bronchitis. Conversely, bioaerosols with particle sizes smaller than

5 μm can directly infiltrate the alveoli, leading to the development of various lung diseases. As bioaerosols traverse the respiratory system, they have the potential to deposit in different sites of the respiratory canal, causing lesions or other forms of damage. Recognizing the potential harm posed by pathogenic bioaerosols is crucial, prompting the implementation of measures to mitigate exposure and protect respiratory health [2,3].

Bioaerosols play a vital role in human health as they are a major source of atmospheric transmission of pathogens, which are often associated with pandemic outbreaks and rapid disease spread, like the Corona Virus Disease 2019 (COVID-19) [4,5]. The rapid spread of COVID-19 throughout the globe has posed an unprecedented challenge to public health, education, and trade systems. The high transmission rate of severe acute respiratory syndrome coronavirus-2 (SARS-CoV-2) and mortality rate of COVID-19 have significantly impacted human society. Infected individuals release atomized particles and pathogens that are heavily suspended on dust-forming aggregates. Floor dust samples from a COVID-19 patient room tested positive for the SARS-CoV-2 [6].The profound impact of biological aerosols on human health is evident from the significant number of respiratory infections and associated complications that occur worldwide each year [7].

Understanding and controlling the spread of pathogens in different environments requires knowledge of their sources, sampling techniques, and monitoring methods. Selecting an effective sampling strategy is crucial for collecting biological aerosols for various research purposes [8]. Passive sampling, such as using petri dishes to collect settled dust, is a simple and low-cost method that provides qualitative measurements of microbial air concentrations [9]. In contrast, active sampling involves drawing a preset volume of air through a pump, allowing airborne microbes to settle on culture media, fine fibers, or porous membrane structures. This method is suitable for microbial sampling of inhalable dust and allows for the quantification of biological aerosol concentrations [10]. Detecting pathogens is vital for controlling and preventing potential deadly disease outbreaks [11]. Bioaerosol detection strategies involve traditional offline and online measurement techniques [12]. However, effective online monitoring remains a challenge due to the complexity and diversity of bioaerosol sampling, as well as the significantly changing spatiotemporal nature of bioaerosols.

Mathematical knowledge and computational models can help predict trends in pathogen transmission, assisting public health authorities in taking pre-emptive measures to prevent disease outbreaks, thereby reducing case numbers and mortality rates [13]. The predictions of these models have played a pivotal role in the public health strategies of many countries [14]. Constructing and applying these models require interdisciplinary collaboration, incorporating knowledge and techniques from fields such as epidemiology, biostatistics, environmental science, and computer science. Continuously improving and updating these models can equip human societies with sufficient knowledge to respond appropriately to possible infectious threats in the future. Summarizing the state-of-the-art prediction models and techniques can highlight research gaps and challenges, stimulate new research ideas, and advance scientific progress in the field of bioaerosols. This, in turn, can provide valuable insights for future public health practices and scientific research.

The use of environmental DNA (eDNA) technology in biodiversity and conservation studies has become increasingly prominent, as demonstrated during the COVID-19 pandemic. By analyzing genetic material in the environment, eDNA technology enables the monitoring and identification of various biological species, including viruses. During the COVID-19 pandemic, this technology was utilized to track and monitor the spread of the virus, particularly in detecting its presence in air samples [15]. This approach has provided a new tool for epidemic surveillance, helping scientists better understand how viruses spread in the environment and how to control this spread effectively. The dual role of eDNA technology in both biodiversity research and disease control highlights its extensive applications and significant value in modern scientific research.

Despite numerous published studies on biological aerosols, there is a lack of comprehensive developmental trend analysis in the field. This review aims to provide a visual bibliometric analysis of bioaerosol research over the past three decades, analyzing emergent

keywords reflecting research hotspots and trends in different eras. Additionally, detailed technical comparisons will be made and key research needs will be identified to drive the field's development in the coming years. This review aims to provide valuable insights for researchers in the field and contribute to the advancement of bioaerosol research, ultimately leading to improved public health practices.

## 2. Materials and Methods

### 2.1. Databases Used

The literature retrieved in this review was mainly from the Web of Science (WoS) core collection database, with "(bioaerosol* OR biological aerosol* OR microbial aerosol* OR microbiological aerosol*)" as the search topic and "Article" and "Review" selected as the search type and the literature set back to the year 1 January 1990 to 31 December 2023. The search yielded 9395 documents (retrieved and downloaded on 6 January 2024), and the records were exported as "plain text files" with "full records and cited references".

### 2.2. Information Analysis System

This review has been conducted with the help of the bibliometric software CiteSpace (5.8.R5), which performs a visual analysis based on keywords, facilitating a review of the development of the entire field of research. The CiteSpace software identifies the research frontiers in a given field using a burst word detection algorithm, which is used to identify the research frontiers of a given field by counting the frequency of words in the titles, abstracts, keywords, and identifiers in the literature of the selected field and, based on the frequency of the word growth to identify research frontier terms, these emergent words are considered as indicators of research frontier topics [16]. The retrieved literature was imported into CiteSpace with the time slice set to 1. The top 20 keywords in terms of emergence intensity were selected for presentation and, according to the keyword emergence table, some of the major research directions in previous years as well as the emergence of emerging fields in recent years can be seen.

### 2.3. Information Detection System

The keyword emergence detection algorithm in CiteSpace software is mainly based on Kleinberg's emergence detection algorithm. Kleinberg's algorithm is an algorithm for detecting emergent events in time series data. Kleinberg's algorithm identifies words or phrases with a significant increase in frequency during a certain period of time, which are considered "bursts" and indicate that these words or phrases have a high level of research interest or social concern during that period of time. Kleinberg's algorithm monitors data streams by constructing an automated state machine model to recognize the occurrence of emergent events. The core statistical method employed in Kleinberg's burst detection algorithm is dynamic programing. Dynamic programing is a method used across various fields such as mathematics, management science, computer science, economics, and bioinformatics for optimizing decision-making processes. Kleinberg's algorithm ingeniously translates the problem of detecting bursty patterns into a problem of finding the optimal path, showcasing the powerful capability of dynamic programing in handling complex decision processes [17]. By quantifying the costs of state transitions and seeking the path with the minimum overall cost, the algorithm effectively identifies and quantifies burst patterns in data, which has significant applications in time series analysis, text mining, and social media analytics.

## 3. Result and Discussion

### 3.1. Analysis of Research Trends

The research literature in the field of bioaerosols demonstrates a consistent upward trend in annual publications, as indicated in Figure S1. This indicates a sustained interest and focus on the field. Since 2010, the number of papers published per year has consistently surpassed 300, with a significant exponential increase in 2019. The surge in papers in

2010 can be attributed to the outbreak of influenza A (H1N1), while the sharp rise in 2019 is closely linked to the emergence of novel coronavirus pneumonia. This increasing trend in research on bioaerosols has garnered significant attention in recent years. The rapid growth in the number of publications signifies that bioaerosols are becoming an important topic in international social science research.

This review analyzes the top 20 keywords in terms of emergence strength over the period 1990–2023, based on the CiteSpace software's algorithm for detecting bursts of words. Figure 1 shows the changes in the hotspots of bioaerosol research over the past 30 years. Based on the emergence of keywords in different periods, we categorize the development of the bioaerosol field into three stages:

**Figure 1.** Top 20 keywords with the strongest citation bursts. Blue line represents the base timeline, and red part indicates the burst duration of each keyword.

Between 1990 and 2010, numerous emergent terms gained prominence in the field of bioaerosol research. Among these terms were "lung", "mice", "aerosol", "endotoxin", "enumeration", "hypersensitivity pneumonitis", "worker", "organic dust", "sampler", "spore", and "symptom".

The focus on "lung" and "hypersensitivity pneumonitis" can be attributed to studies exploring the impact of bioaerosol on respiratory health. Notably, in-depth research on respiratory infectious diseases such as Severe Acute Respiratory Syndrome (SARS) in 2002, H1N1 influenza in 2009, and Middle East Respiratory Syndrome (MERS) in 2012 shed light on the transmission mechanisms of these diseases. These flu viruses can spread through tiny droplets expelled from an infected individual's respiratory tract during sneezing or coughing, forming dispersed viral bioaerosols that can be inhaled by others, thereby facilitating the spread of the virus [18]. As most transmission routes of these diseases are associated with airborne particles and the lungs serve as the primary site for gas exchange in the body, the lungs become susceptible to infection. Furthermore, considering that bioaerosols can carry various allergens such as mold, bacteria, and chemicals, the inhalation of these particulate matters can lead to allergic pneumonitis [19]. Therefore, investigating lung diseases and hypersensitivity pneumonitis in the bioaerosol field holds significant importance, guiding our understanding of disease transmission mechanisms and informing preventive measures.

The emergence of the term "mice" underscores the vital role of animal experimentation in studying the effects of bioaerosol on organisms. Mice have been extensively utilized in biomedical research to examine various infectious diseases, including influenza [20], hepatitis [21], and Ebola virus [22]. Furthermore, mice are particularly valuable for studying human susceptibility and immune responses to specific infections. Through genetic modification, mice possess similar genetic characteristics to humans, enabling researchers to simulate human biological processes in mice when studying the development and treat-

ment of diseases. These characteristics include, but are not limited to, genetic models of human diseases, molecular biology, phenotypic expression, and the simulation of disease symptoms. For instance, humanized mouse models are extensively used for research into infectious diseases, neurodegenerative diseases, and inflammatory diseases by introducing human-specific genes into mice or grafting human organs or cells [23]. These models not only simulate the progression of human diseases but also serve to test the safety and efficacy of new therapeutic methods and preventive strategies.

The attention given to the terms "aerosol", "endotoxin", "spore", "worker", "organic dust", "sampler", and "enumeration" signifies a focus on occupational health, particularly the risks posed to lung health for workers exposed to organic dust in agricultural and industrial settings. Organic dust often contains microorganisms such as bacteria and fungal spores, as well as endotoxins, which can lead to various health issues, including respiratory diseases and allergic reactions. Monitoring and controlling aerosol levels in work environments can effectively mitigate health risks and provide scientific evidence for enhancing occupational safety standards and workplace environmental health measures. Notably, "enumeration" had the longest period of emergence, remaining an active theme from 1994 to 2013. The complexity of bioaerosols, influenced by factors such as climate [24], geography [25], and human activities [26], necessitates long-term and comprehensive research, resulting in the continued focus on this area.

The appearance of these keywords indicates that scholars in the bioaerosol field have increasingly prioritized human health, environmental monitoring, and occupational safety. However, there remains a lack of adequate development of quantitative assessment tools for bioaerosol exposure, and the absence of effective validation hampers the interpretation of exposure outcomes and the accuracy of risk assessments. Moreover, the absence of advanced analytical tools and theoretical support often presents challenges in interpreting data from bioaerosol samples.

In the period from 2008 to 2019, the focus of research shifted towards the detection techniques of bioaerosols. Prominent keywords during this period included real-time PCR, ultrafine particle, Bacillus subtilis, biological particle, quantitative PCR, and biological aerosol particle.

Real-time PCR (RT-PCR) and quantitative PCR (qPCR) are highly precise and efficient techniques used in bioaerosol research for the detection and quantification of DNA samples [27]. RT-PCR, a specific form of qPCR, enables accurate quantification of microbial DNA in air samples by monitoring fluorescent signals in real time during the PCR process. On the other hand, qPCR encompasses various quantitative PCR techniques, not limited to real-time monitoring [28]. These techniques play a critical role in environmental monitoring, disease diagnostics, and food safety testing. The research on bioaerosols allows researchers to promptly and accurately identify the types of airborne microbes in samples, which is vital for understanding pathogen transmission, monitoring air quality, and assessing public health risks. This phase of research addressed the lack of advanced analytical tools and theoretical support in earlier studies.

Starting from 2020, the academic community has placed great importance on the harmful impact of the novel coronavirus on human society. This has led to the emergence of hot research topics such as the coronavirus, airborne droplet transmission, and SARS-CoV-2. These three keywords have continued to persist until 2023, representing the forefront of current academic research.

The term with the highest emergence is "airborne transmission", which is most likely driven by the global outbreak of COVID-19 and the renewed attention placed on the scientific study of bioaerosols. The rapid worldwide spread of COVID-19 has spurred research into understanding the mechanism of airborne viral transmission, making it a focal point in this field. Dumont-Leblond et al. [29] demonstrated the potential for aerosols to transmit the new coronavirus, while Nanehkaranet et al. [30] have found that an asymptomatic individual infected with SARS-CoV-2 is likely to transmit and expectorate the virus, which can remain viable in aerosols for several hours and on surfaces for several

days while still being infectious [31]. Therefore, research on SARS-CoV-2 provides valuable insights for addressing similar public health challenges in the future and has far-reaching implications in practical terms for protecting public health and preventing future outbreaks.

Keywords that are closely related to each other are typically discussed in tandem within research. By utilizing cluster analysis based on the degree of correlation between keywords, trends and concerns within specific research areas can be identified, while also providing an understanding of the interconnections between different topics. In Figure S2, the results of keyword clustering from 1990–2021 are depicted. Each column represents a distinct cluster, with the size of each keyword (circle) indicating its total frequency of occurrence. The transition of color from purple to green to yellow represents the average year of the keyword's occurrence, delineating the progression from early to late. Notably, the yellow color signifies recent research focus. Notably, Figure S2 clearly illustrates that emerging research hotspots include keywords such as COVID-19, SARS-CoV-2, Particulate Matter/PM2.5, size distribution, and diversity. This observation suggests that the field of bioaerosol-air quality interactions may serve as a crucial research direction in the future.

### 3.2. Sources of Bioaerosols

Bioaerosols are formed by atomization of particles and microorganisms from aquatic environments by sea spray or wave action and from terrestrial ecosystems through wind action. Breaking waves entangle air bubbles, resulting in jet drops and films that hold marine organics and sea salt [32,33]. Millions of microorganisms are released into and deposited back into the atmosphere every day, resulting in a dynamic flow of microorganisms between the air, sea, and land. This dynamic circulation provides vital nutrients to the aquatic phytoplankton through deposition [34,35]. Table 1 shows the range of bioaerosol concentrations reported worldwide, reflecting the effect of different environmental conditions on the distribution of bioaerosol concentrations.

**Table 1.** Concentrations of bioaerosols reported worldwide.

| Location | Bioaerosol Types | Range of Bioaerosol Concentrations (CFU/m$^3$) | Average Concentration (CFU/m$^3$) | References |
|---|---|---|---|---|
| | | a. Indoor | | |
| India—school | Fungi | 656~1799 | Cafeteria 1799<br>Classroom 1388<br>Restroom 992<br>Environmental Lab 801<br>Seminar Hall 728<br>Library 656 | [36] |
| | Bacteria | 924~2750 | Cafeteria 2750<br>Restroom 2647<br>Environmental Lab 1998<br>Classroom 1709<br>Seminar Hall 1695<br>Library 924 | |
| Turkey | Fungi | <LOD~1969 | Dormitory 341<br>Dwelling 168<br>Cafeteria 160<br>Office 113<br>Classroom 110<br>Laboratory 99<br>Sport salon 59<br>Restaurant 44<br>Kindergarten 35<br>Primary school 35<br>Library 18 | [37] |
| China—Wastewater treatment plant | Bacteria | / | Sludge thickening house 2390<br>Fine screen 2279 | [38] |
| | Fungi | / | Sludge thickening house 8775<br>Fine screen 5603 | |

Table 1. *Cont.*

| Location | Bioaerosol Types | Range of Bioaerosol Concentrations (CFU/m$^3$) | Average Concentration (CFU/m$^3$) | References |
|---|---|---|---|---|
| | | b. Outdoor | | |
| China—Qingdao | Bacteria | / | Terrestrial Bacteria 33~664 Marine Bacteria 63~815 | [39] |
| | Fungi | / | Terrestrial Fungi 2777 Marine Fungi 66~1128 | |
| China—Xi'an | Bacteria | / | Winter roof 581 Autumn roof 523.5 | [40] |
| | Fungi | / | Winter roof 1234.4 Autumn roof 1318.9 | |
| Colombia—beach | Bacteria | 108~184 | / | [41] |
| | Fungi | 132~220 | / | |
| Poland—Gliwice | Bacteria | / | Winter 57 Spring 305 | [42] |

In outdoor environments, a wide range of sources contribute to the presence of pathogenic bioaerosols, including natural resources as well as human activities. Natural sources, such as water bodies, air, soil, and wildlife, are major contributors. Water is particularly important as it serves as a vital medium for pathogens. Pathogens can rapidly spread and cause diseases through contaminated water sources, impacting the health of both humans and wildlife [43]. Air, being a significant medium in natural resources, enables the long-distance transmission of various pathogens via bioaerosols, affecting human health [44]. Soil and wildlife also play important roles as sources of pathogens. Soil acts as a natural reservoir for various pathogens and spreads them through contact with plants, animals, and humans [45]. Wildlife, serving as hosts and transmitters of pathogens, plays a vital role in the ecosystem of diseases. For instance, the migration and activities of wildlife can result in the spread of pathogens to distant regions [46]. Birds and bats, in particular, can disperse pathogens through feces, saliva, or other secretions. These pathogens can be airborne, forming bioaerosols that people may become infected by inhaling. Additionally, human activities, such as agricultural operations [47], industrial emissions [48], and inadequate urban sewage treatment [49], contribute significantly to pathogen pollution.

Pathogens in outdoor environments pose a considerable risk to public health. Extensive research conducted by the World Health Organization has shown a strong correlation between water pollution and waterborne diseases, including gastroenteritis, dysentery, diarrhea, and viral hepatitis, often associated with the presence of E. coli in contaminated water [50]. These findings highlight the significant impact of water quality on health and the importance of clean water sources in preventing diseases. Currently, researchers utilize a combination of Eulerian and Lagrangian methods to estimate the infection risk associated with airborne particle concentration and distribution. The Monte Carlo model is particularly appropriate for studying infection risks in scenarios where individual behavior is highly random [51]. It is worth noting that, while researchers tend to employ numerical methods to study the mechanisms of airborne pathogen transmission, experimental methods often provide stronger evidence of the possibility of airborne transmission than simple numerical models. Pathogens present in soil can spread to humans through direct contact or the food chain. Studies by Li et al. [52] have shown that interactions between rhizosphere isolates and auxiliary strains are crucial in inhibiting pathogens. Controlling microbial composition to prevent the growth of pathogenic auxiliaries can be an integral part of a sustainable pathogen control strategy. Compared to wild habitats, human-influenced environments exhibit a higher proportion of zoonotic pathogens. Gibb et al. [53] have discovered that zoonoses, such as Ebola, Lassa fever, and Lyme disease, are caused by pathogens transmitted from animals to humans.

Indoor air quality significantly impacts human health and productivity. Microbial contaminants in indoor environments exist in the form of bioaerosols, leading to indoor air pollution (IAP) [54]. IAP is considered one of the top five risks to public health, given that humans spend 80–90% of their time indoors [55], making the risk of indoor pollution much higher than that of outdoor pollution.

Sources of indoor air contaminants are predominantly related to humans [56], pets [57], indoor plants [58], and building environmental characteristics [59]. Humans represent a significant source of indoor bacteria [60], as they release respiratory droplets of various sizes into the environment when speaking, coughing, or sneezing, which may contain a substantial number of bacteria [61]. Research by Maki et al. [62] indicates that pets indeed impact indoor microbial diversity, with animal fur, feces, and fleas potentially contributing to increased indoor air microbial contamination [63]. Ravindra et al. [64] have found that indoor plants can absorb and process indoor pollutants, reducing toxicity, significantly improving indoor environmental comfort and air quality.

Considering examples such as livestock farms and subway construction environments, Staphylococcus aureus has been identified as an occupational pathogen in chicken [65] and pig [66] farms; Passi et al. [67] observed that subway workers are particularly prone to accumulating over 40% of aerosol particles in their lungs. In summary, microbial contaminants in indoor air mainly originate from human activities, pets, indoor plants, and building environmental features. Collectively, these factors influence the microbial composition and concentration of indoor air.

Understanding the sources and pathways of pathogen transmission across different environments is crucial for preventing and controlling infectious diseases outbreaks. Comprehensive monitoring and management strategies are required to reduce the risk of pathogen transmission.

### 3.3. Methods of Sampling

The primary objective of biological aerosol sampling is to obtain representative samples of biological aerosols from the surrounding air. These samples are then collected onto a sampling medium or placed into a container for further analysis [68]. Sampling is a critical process for detecting biological aerosols present in the atmosphere. The composition, density, size, and shape of biological aerosols can vary significantly, which can have a significant impact on the efficiency of the sampling equipment used [69]. When choosing sampling equipment, various factors should be considered, including the availability of the sampler, the volume of air to be sampled, the cost and convenience of the sampler, the collection efficiency for biological aerosols, the objectives of the research, and the climatic conditions during sampling. Table 2 provides a brief comparison of representative bioaerosol sampling methods in the literature and their advantages and disadvantages.

**Table 2.** Representative bioaerosol sampling methods.

| Sampling Methods | Advantages | Disadvantages | Frequency * 2000–2019 | Frequency * 2020–2023 | References |
|---|---|---|---|---|---|
| | | a. Passive sampling | | | |
| Agar settling plate | Low cost. Simple to operate. Rapid culturable bioanalytical methods. | Limited sampling time. Only the culturable bioaerosol fraction was measured. Bias for collecting bioaerosol particles with larger particle sizes. | 0.031 | 0.04 | [70–73] |
| Electrostatic precipitator | Sampled bioaerosols can be measured using a large number of analytical methods. | High initial and operational costs. Lower efficiency for very fine particles. Decreased performance with high-resistance dust. | 0.009 | 0.005 | [74–78] |

**Table 2.** *Cont.*

| Sampling Methods | Advantages | Disadvantages | Frequency * | | References |
|---|---|---|---|---|---|
| | | | 2000–2019 | 2020–2023 | |
| Vacuum, surface swabs, or wipes | Low cost. Fast sampling. | Age of collected dust samples is not available. Collection efficiency may vary depending on the surface material and the wiping/scrubbing pressure applied during collection. | 0.018 | 0.04 | [79] |
| b. Active sampling | | | | | |
| Impingement | Widely used technology. Large amounts of data available. Highly efficient. Use of liquid media overcomes overloading and enumeration problems. | Drying of the liquid medium due to evaporation. Sampling is required for further quantification. Samples may be contaminated after sampling. | 0.03 | 0.015 | [80–82] |
| Impaction | Simple to operate. Low cost. Direct collection of microorganisms. Ideal as a particle size classifier. | Culturable microorganisms only. Possibility of overloading microorganisms in the plate. Strongly affected by wind speed and direction. Drying of the ager surface decreases efficiency. | 0.086 | 0.11 | [83,84] |
| Filtration | Wide range of applications. Simple to operate. Low cost. Size-dependent biological particle collection. Suitable for a variety of enumeration and identification techniques. | Requires sampling for further quantification. Potential for microbial overload at highly contaminated sites. Risk of microbial desiccation due to continuous airflow. Risk of low microbial viability. | 0.101 | 0.13 | [85–87] |
| Cyclone | Reduced rebound and loss of bioparticles. Better recycling and collection. Easy sterilization process. | Drying of liquid media due to evaporation. | 0.012 | 0.015 | [88,89] |

\* The "Frequency" index in this table was calculated by removing all review articles from the searched literature base and randomly selecting 50 articles per year from the period 2000–2023 to identify the sampling methods used in each article.

The passive sampling methods for bioaerosols rely on natural forces like gravity, electrostatic forces, or a combination of these forces to deposit particles onto the collection medium. The collected particles are usually quantified based on the number of colony-forming units (CFU) within a specified area of the settling plates over a specified time period (e.g., CFU/m$^2$/h). This approach is simple and cost-effective [70]. However, as the settling velocity of particles varies with their size and density; smaller, lighter particles can remain airborne for longer periods but the sampling time is limited [71]. These smaller particles can remain suspended indefinitely if the air velocity exceeds their settling rate [72]. As a result, passive samplers are often considered qualitative tools. Additionally, even in enclosed spaces, subtle changes in airflow can occur, making it difficult to calculate CFU per air volume and potentially biasing the collection towards larger bioaerosol particles [73].

Since fresh aerosol particles, including bioaerosols, carry an electrostatic charge, passive devices that utilize electrostatic collection in addition to gravity deposition may have an advantage over passive samplers that utilize only gravity deposition to capture quantities of biological material [74]. Redmann et al. [75] found in practice that the miniaturized electrostatic precipitator (mEP) can provide the same protection from environmentally contagious bioaerosols as the filter-based respirator (N95) without causing unnecessary pressure drops to the wearer, thus facilitating long-term use in an unobstructed breathing configuration. Therkorn et al. [76] found that the Rutgers Electrostatic Passive Sampler (REPS) collects seven times more total microorganisms than passive PTFE filters that rely solely on gravitational particle deposition. However, high initial and operating costs and low efficiency for very fine particles are disadvantages of the electrostatic precipitator [77].

Although the actual air volume sampled is unknown, passive samplers can still indicate the presence and types of bioaerosols. During the COVID-19 outbreak, Angel et al. [78] demonstrated that PDMS-based passive samplers could assess individual exposure to aerosols and droplets containing SARS-CoV-2. This helped in the early detection of potential cases and guided infection control measures in specific locations to prevent community spread. Pan et al. [79] collected surface swab samples from 24 rooms once occupied by COVID-19-positive students to detect SARS-CoV-2 in isolation dormitories. These studies highlight the efficiency of passive sampling in the sampling and detection of SARS-CoV-2.

Passive sampling methods are particularly suitable for long-term monitoring or in environments where electricity is not available. Fluctuation in the composition and concentration of bioaerosol over time are not likely to significantly affect long-term integrated samples, providing a more representative average bioaerosol concentration. Moreover, passive sampling devices ensure viability and cultivability of microbes, without subjecting them to the stresses associated with active sampling devices.

Active sampling methods involve the use of equipment to actively draw in air samples, thereby collecting biological particles within the aerosols. Active sampling techniques can quantify the concentration of biological aerosols. However, the requirement of air propellers and the associated power demands may limit the flexibility of sampling.

The original liquid impinger design was proposed as early as 1947, which led to the development of the classic all-glass impinger (AGI), which operated at 12.5 L/min and collected particles into 20 mL of liquid [80]. The use of liquid media overcame overloading and enumeration problems [81], but the liquid media would dry out due to evaporation [82].

One of the main advantages of using impactors to collect airborne microorganisms, especially culturable microorganisms, is the simplicity and convenience of use; once the sample has been collected, the agar plate can be transferred directly to the incubator without intermediate steps. Culturable microorganisms represent only a small fraction of the total number of microorganisms, so viable but nonculturable microorganisms and inactive microorganisms that may still pose a health risk are missed [83] and the petri dish may be overloaded with microorganisms [84].

Filtration is one of the most common methods for capturing airborne particles, including bioaerosols, because it is convenient and easy to use [85]. Once bioaerosol particles are collected on the filter, they can be eluted into a liquid for subsequent analysis by various techniques. However, in highly contaminated sites, there is a potential for microbial overload [86]. And, under constant airflow, microorganisms can become desiccated, making them less viable [87].

When air containing aerosols is drawn into the sampler, the air creates a rotating flow inside the sampler. Due to centrifugal force, aerosol particles are thrown against the outer wall of the sampler due to their inertia and are captured in the collection vessel, while the cleaner air leaves the sampler through the center. Cyclone samplers reduce the rebound and loss of biological particles [88] but, again, there is drying of the liquid medium due to evaporation [89].

Ramuta et al. [90] have highlighted the effectiveness of active air samplers in monitoring SARS-CoV-2 across various settings. These devices have successfully detected and

sequenced respiratory pathogens in diverse communal environments. However, there are limitations to these methods, such as the inability of passive sampling to determine the age of the samples, while active sampling requires trained operators and electricity, potentially restricting its use in remote areas.

Despite significant research on the types of biological aerosols in outdoor and indoor environments, as well as their sampling methods, studying biological aerosol exposure remains challenging due to the multitude and variability of factors affecting aerosol content [91]. Consequently, the physiological impacts of bioaerosols on humans are still not clearly understood. The variability of occupational sources for biological aerosols exposes workers [92] to health problems and makes it difficult to predict exposure outcomes using applicable methods [93]. Future recommendations include the use of dosimetry and dose–response models to estimate infection probabilities in high-risk populations. Further research is needed to detect anticipated effect sizes over longer time spans and determine the impacts of chronic and acute exposures.

### 3.4. Detection Techniques for Bioaerosols

Detection techniques for bioaerosols can be categorized into offline detection and online detection methods. Offline detection involves collecting a representative bioaerosol sample and conducting intensive laboratory-based analysis. These methods, such as culture colony counting [94], microscopic identification [95,96], and DNA amplification using PCR [97], have limitations in generating high-resolution time-series data required for online methods.

While plate culture colony counting is widely used, low-cost, and easy to perform [98], it is only applicable to culturable microorganisms [99], while nonculturable particles like viruses are investigated using RT-PCR, chip-based digital PCR (dPCR), and droplet digital PCR (ddPCR) [6]. And there is a possibility of contamination of the petri dish, leading to microbial overload affecting the results [100,101]. Microscopy allows observation of microbial morphology and counting of total bioaerosols [102], but operators need specialized skills [103]. PCR is effective in detecting pathogen DNA [104] but demands high purity in samples [105]. These techniques are not suitable for real-time detection due to limitations in sample volume, operational speed, and nondestructive microbial analysis.

In contrast, online detection methods for biological aerosols include mass spectrometry analysis, Raman spectroscopy, flow cytometry, and biosensors. Mass spectrometry identifies the composition, structure, and purity of compounds in various sample types [106], but equipment costs are high and operator skills are demanding [107]. Raman spectroscopy is a rapid and noninvasive technique that provides fingerprint information about microorganisms [108,109]. The amino acid and protein content of a biological aerosol has a significant influence on the spectra generated using Raman spectroscopy [110]. Flow cytometry analyzes scattered light to determine the quantity, size, and shape of microorganisms [111], but flow cytometry requires a high level of sample handling [112]. Biosensors combine biological recognition elements (such as enzymes, antibodies, cells, and DNA) with a physical detection system (such as optical, electrochemical, or mass spectrometry sensors) for detecting chemical substances [113]. Biosensors maximize sensitivity even deprived of molecular amplification raising the prospect of quick investigation of pathogenic agents [114].

During the COVID-19 pandemic, Li et al. developed a novel single-particle mass spectrometry technique for real-time and sensitive detection of bioaerosols [115]. It integrates single-particle aerosol mass spectrometry, Matrix-Assisted Laser Desorption/Ionization Time of Flight Mass Spectrometry (MALDI-TOF MS), and fluorescence spectroscopy and is designed to detect bioaerosols ranging from 0.5 to 10 µm. This technology provides a robust tool for public health monitoring. Pan et al. [116] proposed a microbial amplification detection method that enables the visualization and counting of microbes. This method, unlike conventional chemical and biosensing techniques, enables absolute counting of microbial particles and its simple principle can be adapted into devices suitable for various

living scenarios. Yousefi et al. [117] developed an electrochemical sensor functionalized with kinetically responsive antibody probes, achieving reagent-free sensing of SARS-CoV-2 viral particles and antigens within five minutes. A brief comparison of centrally represented bioaerosol detection techniques is presented in Table 3.

**Table 3.** Representative bioaerosol detection technologies.

| Detection Technologies | Advantages | Disadvantages | Frequency * | | References |
|---|---|---|---|---|---|
| | | | 2000–2019 | 2020–2023 | |
| *a. Offline detection technologies* | | | | | |
| Conventional colony counting and culturing methods | Low cost. Easy to operate. Can be used in dilution methods and selective media for specific microorganisms. | Time-consuming. Contamination problems may be encountered. Microbial overload may affect results. | 0.113 | 0.09 | [98–101] |
| Classical Microscopy | Easy to operate. Microbial morphology can be observed. | Professional skills are required. The information available is limited. | 0.042 | 0.01 | [102,103] |
| Polymerase chain reaction (PCR) | High sensitivity. | Requires specific primers. High sample purity requirements. | 0.055 | 0.125 | [104,105] |
| *b. Online detection methods* | | | | | |
| Mass Spectrometry | Rapid identification. Provides molecular fingerprint. | High equipment costs. High operator skill requirements. | 0.055 | 0.025 | [106,107] |
| Raman spectroscopy | Eliminates the need for complex sample handling. Provides chemical structure information. | Takes longer to obtain data. | 0.004 | 0.01 | [108–110] |
| Flow cytometry | High throughput. Enables rapid analysis of cell size and complexity. | Higher requirements for sample handling. | 0.009 | 0.01 | [111,112] |
| Biosensor | Higher sensitivity and specificity. Highly customizable. | Requires specific biometric elements. May be environmentally sensitive. | 0.004 | 0.015 | [113,114] |

* The "Frequency" index in this table was calculated by removing all review articles from the searched literature base and randomly selecting 50 articles per year from the period 2000–2023 to identify the detection technologies used in each article.

While these advancements in bioaerosol detection technology have enhanced our ability to monitor and respond to the risks of diseases spread by bioaerosols, current detection technologies still have limitations. They may struggle to detect bioaerosols in low concentrations or complex samples and their reliability can be affected by environmental conditions. The high cost and technical expertise required for their operation also limit their widespread application in diverse settings. In general, the use of bioaerosol monitoring techniques may vary depending on factors such as geographic location, local environmental policies, technology availability, research priorities, and economic conditions. For example, in countries and regions with relatively limited economic resources, traditional sampling and laboratory analysis methods are often preferred due to their relatively low cost and technical requirements. These methods do not require highly sophisticated equipment or techniques, making them more feasible for limited budgets and technology bases.

Different countries and continents have different requirements for regulations and guidelines for environmental protection and public health protection, which affect the

scope and methods of bioaerosol monitoring. For example, the UK has issued guidance on strategies for environmental monitoring of bioaerosols at regulated facilities, proposing the use of culture-based methods and the collection of samples by shock and filtration. This suggests a more structured approach to bioaerosol monitoring in the UK, particularly in environments known to release a wide range of bioaerosols including bacteria, fungi, and viruses, such as sewage treatment plants. In contrast, Poland has not established limit values for airborne microorganisms and endotoxins in workplaces, including sewage treatment plants. This suggests that the regulation of bioaerosol concentrations in some environments is less stringent in Poland and highlights the challenge of establishing harmonized standards for bioaerosol monitoring [118].

Climatic and geographic conditions in different regions can affect the growth of bioaerosols, leading to differences in monitoring strategies. For example, due to the high humidity in the tropics, researchers and scholars may pay more attention to the monitoring of molds and fungi [119], whereas temperate regions are rich in plant species and have four distinct seasons, with spring and fall in particular being the peak time for many plants to bloom and release pollen. This seasonal variation leads to significantly higher pollen concentrations during specific time periods, posing a threat to allergy and asthma sufferers [120] and, thus, in temperate regions, researchers and scholars may be more concerned with pollen monitoring [121].

Despite these differences, there is growing global awareness of the importance of bioaerosol monitoring in public health, environmental protection, and occupational safety. This has led to efforts to standardize methods and share best practices, although significant differences remain.

### 3.5. The Development of Mathematical Models in Pathogen Transmission Prediction

An intricate, often invisible, gradual process of spillover involving animals, livestock, vectors, and the abiotic environment leads to the spread of novel diseases among human populations. Developing mathematical models to predict pathogen transmission is crucial for ecological epidemiologists to address the growing threat of infectious diseases. These models utilize mathematical approaches and consider the specificity of host–pathogen or host–vector interactions. They serve as tools for analyzing the dynamics of infectious diseases and potential control strategies [122].

After aerosolization, bioaerosols disperse and travel along with ambient particulate matter over long distances before eventually being deposited. The diffusion and transport pathways of bioaerosols can be categorized into three stages: launching, transport, and deposition. Bioaerosols can be launched through air turbulence [123], fungal spore discharging [124], water or wind action over contaminated surfaces [118], and other similar mechanisms. The dispersion of bioaerosols in the atmosphere is influenced by time and distance [125], and it can be divided into submicroscale, microscale, mesoscale, and macroscale. Submicroscale transport occurs in short periods of time (<10 min) and short distances (100), typically observed in buildings and narrow areas. The most common type of transport is microscale, which spans between 10 min and 1 h and distances of 100 m to 1 km. Mesoscale transport refers to transportation over the span of days and distances up to 100 km, while macroscale transport extends to even longer durations and distances. Bioaerosols are ultimately deposited through gravitational settling, surface impacts, downward diffusion of molecules, rainfall, and static electricity. Aerosol pathogens can remain in the air for extended periods and be transmitted over long distances, leading to rapid and widespread disease transmission [126].This is especially a concern in enclosed or poorly ventilated environments [127]. Pathogens transmitted through aerosols, such as SARS-CoV-2, can cause severe respiratory infections, increasing the risk of severe illness and death. Therefore, it is crucial to model the spread of pathogens through aerosols in order to predict and mitigate their transmission.

In 1927, Kermack et al. [128] introduced the influential Susceptible-Infected-Removed (SIR) model, focusing on fundamental dynamics of infectious diseases such as transmis-

sion rates, recovery rates, and immunity. This model laid the groundwork for modern epidemiological modeling and greatly influenced subsequent research. Despite their importance, early models faced challenges in accurately capturing the actual process of pathogen transmission, as they often relied on idealized assumptions and were limited by the lack of detailed epidemiological data.

In the 2000s, prediction models began integrating more complex dynamics and fluid mechanics elements. Liu et al. [129] utilized the three-dimensional Reynolds-averaged Navier–Stokes equations and the RNG k-epsilon turbulence model to simulate airflow fields. By assuming no secondary infections, they derived pathogen concentrations around buildings over time and identified infection zones based on known pathogen concentrations. These studies significantly advance the prediction and control of airborne viruses.

Gloster et al. [130] summarized the results of a 2008 workshop held at the Institute of Animal Health (UK), comparing models of windborne transmission and infection of foot-and-mouth disease (FMD). Despite significant uncertainties in input parameters (virus release, environmental fate, and subsequent infection), the workshop demonstrated that, under favorable meteorological conditions, the risk of long-distance infection could not be ignored. These studies contribute to progress in predicting and controlling airborne viruses.

Between 2010 and 2019, research increasingly focused on aerosol transmission in indoor environments, particularly medical facilities and public spaces. Key considerations included the influence of air circulation, humidity, and temperature on aerosol spread. Seo et al. [131] incorporated Particle Number Concentration (PNC) and meteorological conditions into predictive models, assessing accuracy using Mean Absolute Percentage Error (MAPE) to forecast microbial contamination in indoor environments. Kulkarni et al. [132] used the Sulfur Transport and Emission Model (STEM) chemical transport model and the Weather Research and Forecasting (WRF) meteorological model to analyze particulate matter concentration, seasonal cycles, and contributions from different sources and regions in Central Asia. Their simulations for 2030 indicated increasing PM2.5 and black carbon (BC) levels in the absence of emission control, potentially exacerbating human and ecological health impacts and climate warming. These models provide timely recommendations for environmental pollution control.

In response to the severe consequences of the COVID-19 outbreak, aerosol transmission models have gained increased attention since 2020. Ho et al. [133] employed Computational Fluid Dynamics (CFD) models to simulate spatiotemporal variations in airborne pathogen concentration and estimated infection risks at different distances and exposure times. Their findings highlighted the significant influence of air circulation [134], pathogen concentration in the air, and mask usage [135] on transmission and exposure risk [136]. Guo et al. [137] obtained the spatial distribution of the probability of infection (PI) by combining the Spatial Flow Influence Factor (SFIF) approach, which makes full use of space and equipment in the built environment, with the Wells–Riley model. It is important for controlling the spread of COVID-19 and other infectious respiratory diseases, as well as for promoting the development of sustainable cities and societies. However, due to the complex and variable nature of virus transmission and the factors influencing it, certain assumptions and parameters in the models might not fully reflect the actual situation [138]. Moreover, the application of these models is limited by the quality and availability of data. Inaccurate or incomplete data can compromise the accuracy of model predictions.

Overall, these studies offer crucial theoretical support for understanding and assessing the risk of airborne diseases. However, enhancing the accuracy and reliability of models in practical applications requires further data collection and model improvements.

### 3.6. Application of eDNA Technology on Bioaerosol Study

eDNA technology, which refers to the degraded genetic material shed from organisms [139], has been widely used in research. It involves two approaches known as barcoding and metabarcoding [140]. Barcoding uses species-specific primers to detect DNA fragments of a single species within a sample [141], while metabarcoding uses universal primers to detect

millions of DNA fragments from various species [142]. This technique has been commonly used in aquatic ecosystems to assess biodiversity, such as in fish, amphibians [143], and bacterial antibiotic resistance genes (ARGs) [144]. Terrestrial ecosystems have also been assessed using eDNA, including terrestrial vertebrate diversity and global plant diversity using soil eDNA and pollen DNA [145,146]. However, the application of eDNA technology in bioaerosol studies lags behind due to several challenges.

One challenge is the low microbial composition in air samples, which can affect the detection limits and sensitivity of downstream genomic analysis [147]. These analyses typically require sufficient quantities of DNA for accurate analysis [148]. Additionally, there is some debate regarding whether eDNA includes both intracellular DNA (iDNA) and extracellular DNA (exDNA) or only extracellular DNA [149]. Despite this, researchers have used eDNA to characterize bioaerosols using quantitative polymerase chain reaction (qPCR) and next-generation sequencing (NGS).

The primary objective of eDNA technology is to monitor and assess biodiversity. It involves identifying and enumerating species by analyzing DNA extracted from environmental samples, which is particularly useful for tracking non-observable and endangered species. In contrast, traditional bioaerosol sampling techniques focus on collecting biological particles in the air for environmental monitoring, health risk assessment, and disease prevention. While there are differences in aims and analysis steps between eDNA technology and traditional techniques, there is some overlap in sampling methods.

In terms of monitoring techniques and data analysis, eDNA employs molecular marker technologies like PCR and metabarcoding [150], while traditional techniques emphasize microbiological culturing, microscopic identification, and biochemical analysis. The use of eDNA technology allows for more accurate detection of target species, especially in cases of low species density [151]. It also eliminates the labor-intensive nature and reliance on morphological taxonomic expertise by combining high-throughput sequencing with bioinformatics analysis to simultaneously detect multiple taxa and species.

Both eDNA technique and traditional bioaerosol sampling face their own technical challenges, but there are opportunities for mutual benefit. For instance, the noninvasive sampling methods and efficient molecular detection technologies of eDNA can be applied to the analysis of bioaerosol samples, enhancing sensitivity and accuracy. Conversely, techniques used for capturing and analyzing airborne microorganisms in bioaerosol sampling can aid in the treatment and analysis of eDNA samples, particularly when assessing airborne biodiversity. With advancements in molecular biological techniques, these methods may have more crossover applications in the future.

Currently eDNA techniques are less commonly used in the study of bioaerosols, but recent studies have started applying these techniques to target bacteria, fungi, and plants in air samples. Harnpicharnchai et al. [152] utilized a sampler called AirDNA, which allows for the collection of high yields of bioaerosols genomic DNA in a short period of time, making it ideal for monitoring the air in built environments and for health and environmental studies. Banchi et al. [153] assessed the taxonomic composition plus diversity of airborne fungi and compared the molecular data with those obtained by conventional microscopy in a study and found that macro-barcoding shows a tenfold efficiency over conventional microscopic analysis on identifying fungal taxa, confirming its technical performance. Soon after, Banchi et al. [154] successfully applied a combination of eDNA macro-barcoding and high-throughput screening (HTS) techniques to simultaneously detect and characterize faunal and fungal diversity on air using multiple primer combinations. However, the detection and characterization process remains challenging due to the diversified taxonomic groups and different relative abundances found in airborne samples.

In conclusion, while eDNA technology has been successfully applied in various ecosystems, its application in bioaerosol studies is still limited. Overcoming technical challenges and exploring the potential synergies with traditional sampling techniques can lead to further advancements in the field. With the continuous development of molecular

biological techniques, the future holds promise for crossover applications of eDNA and bioaerosol sampling methods.

## 4. Conclusions

Research in the field of bioaerosols has been conducted worldwide over the past three decades, with future directions primarily focusing on the airborne transmission of coronaviruses, the impact of environmental factors on disease transmission, and the effects of air pollution and particulate matter size on health.

Between 1990 and 2023, the research focus in the bioaerosol field underwent significant shifts. Initially, it concentrated on the impact of bioaerosols on respiratory health, reflecting public health concerns regarding the influence of environmental factors on human well-being. As technology advanced and our understanding of bioaerosols deepened, the focus gradually shifted towards the development of more precise and efficient detection techniques. In recent years, with the outbreak of global pandemics, research emphasis has further evolved, specifically on viral bioaerosols and their implications for human society. This shift signifies a move from fundamental biomedical research to more applied and societal-level studies, highlighting the growing importance of global health security. These transitions in the bioaerosol field demonstrate the dynamic nature of scientific inquiry and the role played by societal needs and global health challenges.

This review provides insights into the sources of pathogenic bioaerosols and their potential impact on human health, revealing the complexity and diversity of microbial contamination in both outdoor and indoor environments. In the outdoor environment, natural resources and human activities work together as a significant source of pathogenic bioaerosols, and this interaction highlights the far-reaching impact of the human–nature relationship on public health. Microbial contaminants in the indoor environment, on the other hand, originate mainly from human activities, pets, indoor plants, and features of the built environment, and these sources reflect the complexity of the microecology of indoor spaces and their direct impact on the health of occupants. In addition, this review also emphasizes the importance of monitoring and controlling pathogenic bioaerosols in the indoor and outdoor environments, as well as the adoption of effective strategies to reduce the risks posed by these contaminants to human health.

Over the past three decades, both before and after the emergence of COVID-19, the scientific work primarily employed impaction and filtration sampling methods, followed by colony counting and PCR-based detection techniques. Air samples are collected into liquid in filters, allowing for the continuation of subsequent procedures like DNA/RNA extraction. This makes the agar-based impactor convenient in that, after a sample is taken, the agar plates are transported straight to an incubator without the need for any intermediate stages. According to the targets of any research, supporting PCR-based offline detection with the preferable online detection instruments at a time would maximize the quality and precision of the study outcome. While current research on the types and sampling techniques of bioaerosols in different environments is relatively mature, the continuous changes in factors affecting bioaerosol content present significant challenges for studying bioaerosol exposure. To address this gap, future research should focus on the application of dosimetry and dose–response models to predict infection risk more accurately. Moreover, more extensive studies are necessary to evaluate expected effect levels and clarify the long-term impact of bioaerosol exposure on human health.

Prior to the outbreak of COVID-19 in 2019, predictive modeling in epidemiology, while foundational, was often constrained by idealized assumptions and a lack of data. However, starting from 2020, the focus of research shifted towards more detailed studies of indoor aerosol transmission, employing advanced fluid dynamics and meteorological conditions. This shift significantly enhanced the accuracy and applicability of predictive models. Such progress not only fostered a deeper understanding of pathogen transmission mechanisms but also provided crucial support for the development of effective public health strategies. Although current research provides a theoretical foundation for understanding

and assessing the risk of airborne diseases, improvements are needed in the accuracy and reliability of practical model applications. The complexity and changing nature of virus transmission mechanisms often results in model assumptions and parameters that may not fully reflect reality. Furthermore, model effectiveness is limited by the quality and availability of data, as inaccurate or incomplete data directly impact prediction accuracy. Therefore, enhancing the practical value of these models requires in-depth data collection and model improvement to ensure they more realistically reflect actual conditions of virus transmission.

In bioaerosol research, eDNA technology is primarily used for biodiversity monitoring and assessing antibiotic resistance genes in the environment. Although research in this area is still in its early stages, it is rapidly progressing and showing great potential. It is expected that eDNA technology will contribute to more critical research and play a progressively significant role in bioaerosol studies in the future.

This review provides information on the sources and detection strategies of pathogens in a variety of ecosystems, advances and inherent limitations of a range of models for predicting aerosol-mediated spread of pathogens, and provides a comparative analysis of the eDNA technique and traditional bioaerosol analysis techniques. However, there are several limitations to consider. First, our literature search, which was largely limited to English-language literature, may have missed important studies in other languages, which may affect the comprehensiveness of our conclusions. Second, our review is based on reviews and articles in the Web of Science (WoS) core collection database from 1 January 1990 to 31 December 2023 and, in view of the rapid development of this field of research, new studies may have been published and these recent findings were not included in our review analysis, which may affect the accuracy of the review.

Based on the conclusions drawn from this review, we believe that future research directions will focus on optimizing strategies to combat airborne droplet transmission of SARS-CoV-2, dosimetry and dose–response modeling, long-term human health effects of exposure to bioaerosols, collection of more extensive virus transmission data, and modeling improvements, extensive viral transmission data and modeling improvements, and application of eDNA technology in the field of bioaerosols.

**Supplementary Materials:** The following supporting information can be downloaded at: https://www.mdpi.com/article/10.3390/atmos15040404/s1, Figure S1: Annual distribution of publications in the field of bioaerosols from 1990 to 2023; Figure S2: Keywords clustering map from 1990 to 2021.

**Author Contributions:** Conceptualization, X.D. and H.X.; methodology, C.Z. and Y.W.; validation, M.Y., M.Y., L.W., W.W. and Z.M.; formal analysis, J.Z., L.T., B.Z., T.G. and M.H.; data curation, C.Z.; writing—original draft preparation, C.Z. and T.G.; writing—review and editing, Y.M. and H.X.; visualization, C.Z.; supervision, H.X.; funding acquisition, H.X. All authors have read and agreed to the published version of the manuscript.

**Funding:** This research was funded by National Natural Science Foundation of China (No. 21976171), Guangxi key research and development program (GuikeAB21220063), and Department of Agriculture and Rural Affairs of Zhejiang Province (2022SNJF055).

**Institutional Review Board Statement:** Not applicable.

**Informed Consent Statement:** Not applicable.

**Data Availability Statement:** The data presented in this study are available within the article.

**Conflicts of Interest:** The authors declare no conflicts of interest.

## Abbreviations

| | |
|---|---|
| COVID-19 | Corona Virus Disease 2019 |
| SARS-CoV-2 | severe acute respiratory syndrome coronavirus-2 |
| eDNA | environmental DNA |
| WoS | Web of Science |
| SARS | Severe Acute Respiratory Syndrome |

| MERS | Middle East Respiratory Syndrome |
| --- | --- |
| RT-PCR | Real-time PCR |
| qPCR | Quantitative PCR |
| IAP | indoor air pollution |
| CFU | colony-forming units |
| mEP | miniaturized electrostatic precipitator |
| REPS | Rutgers Electrostatic Passive Sampler |
| AGI | all-glass impinger |
| dPCR | digital PCR |
| ddPCR | droplet digital PCR |
| MALDI-TOF MS | Matrix-Assisted Laser Desorption/Ionization Time of Flight Mass Spectrometry |
| PCR | Polymerase chain reaction |
| SIR | Susceptible-Infected-Removed |
| FMD | foot-and-mouth disease |
| PNC | Particle Number Concentration |
| MAPE | Mean Absolute Percentage Error |
| STEM | Sulfur Transport and Emission Model |
| WRF | Weather Research and Forecasting |
| BC | Black Carbon |
| CFD | Computational Fluid Dynamics |
| ARGs | antibiotic resistance genes |
| iDNA | intracellular DNA |
| exDNA | extracellular DNA |
| NGS | next-generation sequencing |
| HTS | high-throughput screening |

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
