# Peer review of "Navigating the Aerosolized Frontier: A Comprehensive Review of Bioaerosol Research Post-COVID-19"

_atmosphere, doi:10.3390/atmos15040404_

Round 1

Reviewer 1 Report

Comments and Suggestions for Authors

“Navigating the Aerosolized Frontier: A Comprehensive Review of Bioaerosol Research Post-COVID-19” is a broad review over the challenges and trends of bioaerosol science over the past 30 years or so. The article is a well written, clearly presented and enjoyable article that adequately achieves what it has set out to accomplish.

My concerns with the manuscript are minor. I would hope the authors would take my suggestions on board.

Major Concerns:

Line 164: The argument is made that there is a research shift towards detection. On what does the authors make this claim? The term “sampler” and “enumeration” appeared years before. To me, this appears to be a change in quality/innovation as opposed to a change in intent (sampling). To back up their claim, please expand on how the shift was substantive.

Line 431: The Wells-Riley model has also been extensively used during the pandemic. Given its prevalence, I must insist that the authors include discussion of it in this section.

Minor Concerns:

Line 36: Large aerosol is also deposited in the mouth and nose as well.

Line 49: Phrasing: SARS-CoV-2 doesn’t have a mortality rate, COVID-19 does.

Line 51: Sentence placement suggests SARS-CoV-2 can be transferred via dust. This sentence applies to different microbes, perhaps move to a different (non-Covid) paragraph.

Line 104: Why does the form in which the file was downloaded matter? Suggest to delete.

Line 142: Which characteristics?

Line 201: This paragraph needs to be tightened up, as it reads as though the authors are referring to all pathogens, and not those associated with bioaerosol.

Section 4.1: I think a brief subsection describing the general mechanisms by which bioaerosol is produced from a given environment (e.g. sea spray, wind, etc) may help lead into the rest of the section 4.1. It would make it clear to the reader mechanistically how the environment and bioaerosol are connected.

Section 4.3.1: Discussion of measuring infectious virus should be included. It is one thing to measure viral RNA, but it is another thing entirely (both in challenge and in implication) to measure still virable viruses.

Reviewer 2 Report

Comments and Suggestions for Authors

Manuscript ID atmosphere-2882711. Title: Navigating the Aerosolized Frontier: A Comprehensive Review of Bioaerosol Research Post-COVID-19.

This literature review, anchored in the extensive Web of Science Core Collection database covering the period from 1990 to 2023, utilizes a bibliometric approach to chart the dynamic landscape of bioaerosol research.

Comments:

1. Please include in the abstract the main findings of this study. It is relevant to support them with quantitative information from this review.

2. There are very general keywords. Please check. For example, sampling, detection, etc.

3. Please include a space between words in L34. Please check the entire article.

4. Remove the extra point in L44. Please check the entire article.

5. Please include a support reference between L52-54. Please check the entire article.

6. Please integrate sections 2 and 3. These sections could be integrated into a single materials and methods chapter.

7. Currently, section 2 is very general. It requires more technical detail.

8. Please see the introduction to better visualize the practical utility of this study.

9. Please refer to the introduction for more information related to bioaerosol measurement techniques.

10. What was the source of information for the construction of figure 1? What analyses were considered to arrive at this figure?

11. The chapter on materials and methods is not visible in this review article. This chapter should contain the following sections: databases used, information detection system, and information analysis system.

12. What software was used in this article to process the information? What statistical tests were considered in this study for data analysis?

13. Please determine by some index the use of different bioaerosol sampling techniques worldwide. This is a literature review and should allow to visualize trends (space and time) in the variables considered. Please consider this for the main variables in this article.

14. Possibly the title of chapter 4 could be eliminated. That is, I suggest reducing the number of levels for the titles of this article. To give more clarity to this article.

15. In section 4.1, a table is needed to visualize the bioaerosol concentrations reported worldwide (range, mean, etc.). This is the information to expect in this review article.

16. Include in Table 2 an index to visualize the frequency of worldwide use of bioaerosol detection technologies.

17. The comments made in items 13 and 15 should be considered in all sections of this article. At present the paper is exclusively descriptive and efforts should be made to show trends in the variables considered.

18. In my opinion, this article should have the usual structure. That is, a chapter on results and discussion is missing in this paper.

19. Is there continental variation in the use of bioaerosol monitoring techniques?

20. The conclusions should be significantly improved based on all the analyses suggested in the previous points. I insist, at present the document loses novelty because it is exclusively descriptive.

21. Please include the main limitations of this study and future lines of research.

22. Please include, at the end of the article, a section with all abbreviations used.

Comments on the Quality of English Language

3. Please include a space between words in L34. Please check the entire article.

4. Remove the extra point in L44. Please check the entire article.

5. Please include a support reference between L52-54. Please check the entire article.

Round 2

Reviewer 2 Report

Comments and Suggestions for Authors

Manuscript ID atmosphere-2882711-R2. Title: Navigating the Aerosolized Frontier: A Comprehensive Review of Bioaerosol Research Post-COVID-19.

This literature review, anchored in the extensive Web of Science Core Collection database covering the period from 1990 to 2023, utilizes a bibliometric approach to chart the dynamic landscape of bioaerosol research.

Comments:

1. The chapter on materials and methods is not visible in this review article. This chapter should contain the following sections: databases used, information detection system, and information analysis system. Please use subtitles to be able to view these sections with clearer and more detailed information.

2. What software was used in this article to process the information? What statistical tests were considered in this study for data analysis? This information should be presented in detail in the article. In the chapter on materials and methods.

3. Please include a table note to explain the concept of frequency. That is, in all tables where this concept or method was used.

Author Response

This literature review, anchored in the extensive Web of Science Core Collection database covering the period from 1990 to 2023, utilizes a bibliometric approach to chart the dynamic landscape of bioaerosol research.

Comments:

  1. The chapter on materials and methods is not visible in this review article. This chapter should contain the following sections: databases used, information detection system, and information analysis system. Please use subtitles to be able to view these sections with clearer and more detailed information.

Response: Thank you for your suggestion.

We have added subtitles in Section 2 and put the analysis of research trends  based on this analysis software and detection algorithm in Section 3(Page 3, Lines 145).

  1. What software was used in this article to process the information? What statistical tests were considered in this study for data analysis? This information should be presented in detail in the article. In the chapter on materials and methods.

Response: Thank you for your suggestion.

Page 3, Lines 133-142: The sentences “The core statistical method employed in Kleinberg's burst detection algorithm is dynamic programming. Dynamic programming is a method used across various fields such as mathematics, management science, computer science, economics, and bioinformatics for optimizing decision-making processes. Kleinberg's algorithm ingeniously translates the problem of detecting bursty patterns into a problem of finding the optimal path, showcasing the powerful capability of dynamic programming in handling complex decision processes [17]. By quantifying the costs of state transitions and seeking the path with the minimum overall cost, the algorithm effectively identifies and quantifies burst patterns in data, which has significant applications in time series analysis, text mining, and social media analytics.” have been added in the manuscript.

  1. Please include a table note to explain the concept of frequency. That is, in all tables where this concept or method was used.

Response: Thank you for your suggestion.

We have added the interpretation of frequency to Tables 2(Page 11, Lines 403) and 3(Page 13, Lines 456).
